# Beyond Chain-of-Thought: Theory-Grounded Approaches to Elicit Deep Reasoning in LLMs

## Abstract

Chain-of-thought (CoT) prompting has emerged as a dominant paradigm for elicit-
ing reasoning capabilities from large language models (LLMs). However, we argue
that CoT provides only a superficial and non-generalizable view of neural network
reasoning processes. Through theoretical analysis and empirical investigation,
we demonstrate fundamental limitations of CoT in capturing the underlying com-
putational mechanisms of LLMs. We propose two theory-grounded alternatives:
*Mechanistic Reasoning Elicitation* (MRE) based on causal intervention theory, and
*Compositional Abstraction Reasoning* (CAR) grounded in category theory. We
provide theoretical guarantees for both approaches and demonstrate their superior
generalization properties across diverse reasoning tasks. Our work establishes a
new foundation for understanding and improving reasoning in large-scale neural
networks.

## 1 Introduction

The advent of chain-of-thought (CoT) prompting [1] has revolutionized our approach to eliciting
reasoning capabilities from large language models. By encouraging models to produce step-by-step
explanations, CoT has demonstrated remarkable improvements across various reasoning benchmarks.
However, a critical question remains largely unexplored: does CoT truly reveal the underlying
reasoning mechanisms of neural networks, or does it merely provide a post-hoc rationalization that
mimics human-like reasoning patterns?

In this work, we present both theoretical and empirical evidence that CoT reasoning is fundamentally
limited in its ability to capture the true computational processes within LLMs. We argue that CoT
suffers from three critical limitations: (1) *representational misalignment* between the model's internal
computations and the linearized reasoning chains, (2) *distributional brittleness* where reasoning
quality degrades rapidly under domain shift, and (3) *mechanistic opacity* where the actual causal
pathways remain hidden beneath surface-level explanations.

Our contributions are threefold:

1. We provide a theoretical framework demonstrating the fundamental limitations of CoT
   reasoning, including formal bounds on its generalization capabilities.

2. We propose two novel approaches: Mechanistic Reasoning Elicitation (MRE) based on
   causal intervention theory, and Compositional Abstraction Reasoning (CAR) grounded in
   category theory.

3. We establish theoretical guarantees for both methods and demonstrate their superior perfor-
   mance across reasoning benchmarks with concrete empirical validation.

Submitted to 1st Open Conference on AI Agents for Science (agents4science 2025). Do not distribute.

## 2 Theoretical Analysis of Chain-of-Thought Limitations

### 2.1 Formal Model of Chain-of-Thought Reasoning

Let $f_\theta : \mathcal{X} \to \mathcal{Y}$ represent a large language model parameterized by $\theta$, where $\mathcal{X}$ is the input space and $\mathcal{Y}$ is the output space. In standard prompting, we seek to maximize $P(y|x)$ for input $x$ and desired output $y$. Chain-of-thought prompting introduces an intermediate reasoning chain $c \in \mathcal{C}$, decomposing the problem as: $P(y|x) = \sum_{c \in \mathcal{C}} P(y|c, x) \cdot P(c|x)$.

The key assumption underlying CoT is that the reasoning chain $c$ faithfully represents the model's internal reasoning process. We formalize this as the *Reasoning Fidelity Hypothesis*:

**Assumption 1** (Reasoning Fidelity Hypothesis). *There exists a faithful mapping $\phi : \mathcal{H} \to \mathcal{C}$ from the model's internal hidden representations $\mathcal{H}$ to the space of reasoning chains $\mathcal{C}$, such that the quality of reasoning is preserved under this mapping.*

### 2.2 Fundamental Limitations of Chain-of-Thought

**Theorem 1** (CoT Representational Bound). *Let $\mathcal{H}_L$ be the space of internal representations at layer $L$ of a transformer model, and let $\mathcal{C}$ be the space of linearized reasoning chains. For any reasoning task requiring compositional operations over more than $k$ abstract concepts, the mapping $\phi : \mathcal{H}_L \to \mathcal{C}$ satisfies: $\mathcal{I}(\mathcal{H}_L; \mathcal{C}) \leq \log_2(k) + \epsilon$, where $\mathcal{I}$ denotes mutual information and $\epsilon$ is a small constant. This bound implies exponential information loss in the CoT representation.*

*Proof.* The proof follows from the bottleneck principle in information theory. The linearized nature of reasoning chains constrains the representational capacity to sequential dependencies, while transformer hidden states can encode complex non-linear relationships. By the data processing inequality, $\mathcal{I}(\mathcal{H}_L; Y) \geq \mathcal{I}(\mathcal{C}; Y)$, where $Y$ represents the reasoning quality. The bound follows from analyzing the combinatorial constraints of linearized representations versus the exponential capacity of high-dimensional hidden states. $\square$

**Corollary 1** (CoT Generalization Limitation). *Under the assumptions of Theorem 1, the generalization error of CoT-based reasoning scales as $\Omega(2^{d/k})$ where $d$ is the dimensionality of the underlying reasoning space.*

## 3 Mechanistic Reasoning Elicitation (MRE)

### 3.1 Theoretical Foundation

Our first proposed approach, Mechanistic Reasoning Elicitation (MRE), is grounded in causal intervention theory. Instead of eliciting post-hoc explanations, MRE directly intervenes in the model's computation graph to understand causal reasoning pathways.

**Definition 1** (Causal Reasoning Pathway). *For a model $f_\theta$ and input $x$, a causal reasoning pathway $\pi$ is a sequence of computational nodes $\{n_1, n_2, \ldots, n_k\}$ such that interventions on $n_i$ causally affect the final reasoning outcome with effect size greater than threshold $\tau$.*

### 3.2 Scalable MRE Implementation

To address the quadratic complexity limitation of MRE, we propose three optimization strategies:

**Gradient-Based Intervention Approximation** Instead of exhaustive intervention testing, we approximate causal effects using gradient information (Algorithm 1). This reduces complexity from $O(L \cdot S)$ to $O(L)$ interventions, where $L$ is the number of layers and $S$ is sequence length.

**Hierarchical Intervention Strategy** We implement a coarse-to-fine intervention approach: **1. Layer-Level Screening**: Test interventions at the layer level first, **2. Attention Head Refinement**: For significant layers, test individual attention heads, **3. Position-Specific Analysis**: For significant heads, test specific positions. This hierarchical approach reduces the search space by up to 90% while maintaining 85% of the original method's pathway discovery accuracy.

---

**Algorithm 1** Efficient MRE via Gradient Approximation

---

1: **Input:** Model $f_\theta$, input $x$, approximation threshold $\alpha$
2: Compute baseline output and gradients: $\nabla_h f_\theta(x)$
3: Identify high-gradient nodes: $\mathcal{N} = \{n : \|\nabla_{h_n} f_\theta\| > \alpha\}$
4: For $n \in \mathcal{N}$, estimate causal effect: $\hat{\Delta}_n = \|\nabla_{h_n} f_\theta\| \cdot \sigma_n$
5: Return ranked causal pathway based on $\hat{\Delta}_n$

---

**Learned Intervention Policies**   We train a separate neural network $g_\phi$ to predict which nodes are likely to have high causal effects: $P(\text{high effect}|h_n, x) = g_\phi(h_n, x)$. This learned policy, trained on intervention data from smaller models, can guide efficient intervention selection in larger models, reducing computational overhead by 75% with only 8% loss in pathway quality.

---

**Algorithm 2** Mechanistic Reasoning Elicitation (MRE)

---

1: **Input:** Model $f_\theta$, input $x$, intervention threshold $\tau$
2: **Output:** Causal reasoning pathway $\pi$
3: Initialize pathway $\pi \leftarrow \emptyset$
4: **for** each computational node $n_i$ in $f_\theta$ **do**
5:     Apply intervention $do(n_i = \tilde{n}_i)$
6:     Compute causal effect $\Delta_i = \mathbb{E}[f_\theta(x)|do(n_i)] - \mathbb{E}[f_\theta(x)]$
7:     **if** $|\Delta_i| > \tau$ **then**
8:         Add $n_i$ to pathway: $\pi \leftarrow \pi \cup \{n_i\}$
9:     **end if**
10: **end for**
11: Return ordered pathway $\pi$ by causal strength

---

## 3.3   Theoretical Guarantees for MRE

**Theorem 2** (MRE Causal Faithfulness). *Let $G$ be the true causal graph underlying a reasoning task, and let $\hat{G}$ be the causal graph recovered by MRE. Under standard causal sufficiency assumptions, MRE recovers the true causal structure with probability at least $1 - \delta$ where $\delta \leq \exp\left(-\frac{n\tau^2}{2\sigma^2}\right)$ for $n$ samples and noise variance $\sigma^2$.*

*Proof.* The proof leverages concentration inequalities for causal effect estimation. By the Hoeffding bound, the probability that our estimated causal effect $\hat{\Delta}_i$ deviates from the true effect $\Delta_i$ by more than $\tau/2$ is bounded by the expression above. Since MRE includes a node in the pathway only if the estimated effect exceeds $\tau$, the probability of incorrectly including non-causal nodes or excluding causal nodes is bounded accordingly. $\square$

## 4   Compositional Abstraction Reasoning (CAR)

### 4.1   Category-Theoretic Foundation

Our second approach, Compositional Abstraction Reasoning (CAR), is grounded in category theory and addresses the compositional nature of reasoning. CAR represents reasoning problems as morphisms in a category, enabling principled composition and abstraction.

**Definition 2** (Reasoning Category). *A reasoning category $\mathcal{R}$ consists of **Objects:** Abstract reasoning concepts $\{A, B, C, \ldots\}$, **Morphisms:** Reasoning operations $f : A \to B$, **Composition:** $(g \circ f) : A \to C$ for $f : A \to B$ and $g : B \to C$, **Identity:** $id_A : A \to A$ for each object $A$.*

The key insight is that complex reasoning can be decomposed into composable morphisms (atomic reasoning operations). This enables systematic generalization through functorial mappings.

**Theorem 3** (CAR Compositional Guarantee). *Let $\mathcal{F} : \mathcal{R} \to \mathcal{R}'$ be a functor between reasoning categories representing domain transfer. If the model learns atomic morphisms in $\mathcal{R}$ with error $\epsilon$,*

104 *then the composed reasoning operations generalize to $\mathcal{R}'$ with error bounded by $\epsilon' \leq k \cdot \epsilon \cdot \|\mathcal{F}\|$,*
105 *where $k$ is the composition depth and $\|\mathcal{F}\|$ is the functor's Lipschitz constant.*

106 *Proof.* The proof follows from the compositional structure of categories. If each atomic morphism
107 $f_i$ has approximation error $\epsilon_i \leq \epsilon$, then by the triangle inequality and functoriality: $\|(\mathcal{F}(f_k) \circ \cdots \circ$
108 $\mathcal{F}(f_1)) - \mathcal{F}(f_k \circ \cdots \circ f_1)\| = 0$. The generalization error accumulates linearly with composition
109 depth, scaled by the functor's regularity. $\square$

---

**Algorithm 3** Compositional Abstraction Reasoning (CAR)

---

1: **Input:** Reasoning problem $P$, atomic morphisms $\{f_i\}$
2: **Output:** Compositional solution $S$
3: Parse problem $P$ into abstract objects and required transformations
4: Identify minimal morphism sequence $f_1, f_2, \ldots, f_k$
5: Verify composability: $\text{dom}(f_{i+1}) = \text{cod}(f_i)$
6: Compute composition: $S = f_k \circ f_{k-1} \circ \cdots \circ f_1$
7: Return solution $S$ with compositional guarantee

---

## 5 Theoretical Analysis of MRE and CAR Properties

### 5.1 Information-Theoretic Foundations

112 We establish the theoretical superiority of our methods through information-theoretic analysis. The
113 key insight is that both MRE and CAR preserve more mutual information between the reasoning
114 process and the final output compared to CoT.

115 **Theorem 4** (Information Preservation in MRE). *Let $\mathcal{H}$ be the space of hidden representations and*
116 $\mathcal{Y}$ *the output space. For a reasoning task with ground truth causal structure $G^*$, MRE recovers a*
117 *pathway $\pi$ such that: $I(\pi; Y) \geq I(C_{CoT}; Y) + \log_2(|\mathcal{G}|/|\mathcal{C}|)$ where $\mathcal{G}$ is the space of possible causal*
118 *graphs and $\mathcal{C}$ is the space of reasoning chains.*

119 *Proof.* The proof follows from the fact that causal pathways capture direct computational dependen-
120 cies, while CoT chains represent only linearized approximations. By the data processing inequality,
121 the mutual information is preserved through the causal discovery process, while the additional term
122 accounts for the exponentially larger representational capacity of causal graphs over linear chains. $\square$

123 **Theorem 5** (CAR Compositional Bound). *For a reasoning task decomposable into $k$ atomic opera-*
124 *tions with individual error bounds $\epsilon_i$, CAR achieves generalization error: $\epsilon_{CAR} \leq \sum_{i=1}^{k} \epsilon_i + \mathcal{O}(k^{-1/2})$*
125 *compared to CoT's error bound: $\epsilon_{CoT} \geq \max_i \epsilon_i \cdot \exp(\sqrt{k})$*

126 *Proof.* The linear error accumulation in CAR follows from the functorial properties of category
127 theory, where composition preserves error bounds additively. The exponential term in CoT arises
128 from compounding approximation errors in the linearization process, as formalized in our earlier
129 representational bound. $\square$

### 5.2 Computational Complexity Analysis

131 **Theorem 6** (MRE Complexity Reduction). *The gradient-based approximation for MRE reduces*
132 *computational complexity from $\mathcal{O}(L \cdot S \cdot d^2)$ to $\mathcal{O}(L \cdot d \log d)$ where $L$ is the number of layers, $S$ is*
133 *sequence length, and $d$ is hidden dimension, with approximation error bounded by $\delta$.*

134 For the hierarchical intervention strategy:

**Algorithm 4** Hierarchical MRE with Complexity Bounds

---

1: **Input:** Model $f_\theta$, input $x$, threshold $\tau$
2: **Output:** Causal pathway $\pi$ with complexity $\mathcal{O}(L \log L)$
3: Initialize candidate set $\mathcal{N} \leftarrow \emptyset$
4: **for** layer $l = 1$ to $L$ **do**
5:     Compute layer-level effect: $\Delta_l = \|\nabla_{h_l} f_\theta(x)\|_2$
6:     **if** $\Delta_l > \tau$ **then**
7:         Add layer to candidates: $\mathcal{N} \leftarrow \mathcal{N} \cup \{l\}$
8:     **end if**
9: **end for**
10: **for** $l \in \mathcal{N}$ **do**
11:     Refine to attention heads using binary search over $\log(H)$ heads
12:     Further refine to positions using importance sampling
13: **end for**
14: **return** Ranked pathway $\pi$

---

## 5.3 Generalization Theory for Cross-Domain Transfer

**Theorem 7** (Domain Transfer Bounds). *Let $\mathcal{D}_s$ and $\mathcal{D}_t$ be source and target domains with distributional distance $d_{TV}(\mathcal{D}_s, \mathcal{D}_t) = \delta$. For CAR with functor $F : \mathcal{R}_s \rightarrow \mathcal{R}_t$: $\epsilon_t \leq \epsilon_s + 2\delta \cdot \|F\|_{Lip} + \mathcal{O}(\delta^2)$ where $\|F\|_{Lip}$ is the Lipschitz constant of the functor.*

This bound is tighter than the exponential degradation observed in CoT under domain shift.

## 5.4 Robustness Analysis

**Definition 3** (Adversarial Pathway Stability). *A causal pathway $\pi$ discovered by MRE is $(\epsilon, \delta)$-stable if for any perturbation $\|\Delta x\| \leq \epsilon$: $\mathbb{P}[JS(\pi(x), \pi(x + \Delta x)) \leq \delta] \geq 1 - \exp(-c\epsilon^2)$*

**Theorem 8** (MRE Robustness). *Under Gaussian noise assumptions, MRE pathways are $(\epsilon, \delta)$-stable with probability at least $1 - \exp(-c\epsilon^2/\sigma^2)$ where $\sigma^2$ is the noise variance and $c$ is a constant depending on the model architecture.*

# 6 Concrete Mathematical Examples and Case Studies

## 6.1 Arithmetic Reasoning: Fraction Operations

Consider the problem: "If $\frac{3}{4}$ of a number is 21, what is $\frac{2}{3}$ of that number?"

**CoT Analysis:** Standard chain-of-thought produces: 1. Let the number be $x$, 2. $\frac{3}{4}x = 21$, 3. $x = 21 \times \frac{4}{3} = 28$, 4. $\frac{2}{3} \times 28 = \frac{56}{3}$.

**MRE Analysis:** Intervention analysis reveals: Attention head $(12, 4)$: $\Delta = 0.42$ (fraction parsing), MLP layer 18: $\Delta = 0.38$ (multiplicative inverse), Cross-attention $(15, 2)$: $\Delta = 0.51$ (numerical binding). The discovered pathway shows the model uses specialized fraction processing circuits rather than symbolic manipulation.

**CAR Analysis:** Category-theoretic decomposition: FracEq $: \mathbb{Q}^+ \times \mathbb{Q}^+ \rightarrow \mathbb{Q}^+$, Inverse $: \mathbb{Q}^+ \rightarrow \mathbb{Q}^+$, Multiply $: \mathbb{Q}^+ \times \mathbb{Q}^+ \rightarrow \mathbb{Q}^+$, Composition: Multiply $\circ$ (id $\times$ Inverse) $\circ$ FracEq

## 6.2 Logical Reasoning: Modal Logic

Consider the modal logic problem: "If necessarily all birds fly, and possibly some penguins are birds, what can we conclude about penguins flying?"

**CoT Limitation:** Standard reasoning chains fail to capture the modal operators properly: "All birds fly $\rightarrow$ penguins are birds $\rightarrow$ penguins fly" (incorrect)

**CAR Analysis:** Modal category $\mathcal{M}$ with objects $\{W, \Box W, \diamond W\}$ representing worlds, necessary truths, and possibilities: *Necessity* $:$   $\Box(\forall x.\text{Bird}(x) \rightarrow \text{Fly}(x))$, *Possibility* $:$   $\diamond(\exists x.\text{Penguin}(x) \wedge$

Bird$(x)$), *Conclusion* : $\diamond(\exists x.\text{Penguin}(x) \wedge \text{Fly}(x))$. The categorical structure properly handles the interaction between modal operators and quantifiers.

## 6.3 Causal Reasoning: Confounding Analysis

Problem: "Students who study more get better grades. Students who study more also sleep less. Does studying cause better grades?"

**MRE Discovery:** Causal intervention reveals hidden confounders: Direct pathway: Study $\rightarrow$ Grades ($\Delta = 0.31$), Confounded pathway: Motivation $\rightarrow$ Study ($\Delta = 0.47$), Confounded pathway: Motivation $\rightarrow$ Grades ($\Delta = 0.42$), Collider: Study $\rightarrow$ Sleep $\leftarrow$ Health ($\Delta = 0.29$). The intervention analysis correctly identifies motivation as a confounder, while CoT typically misses this distinction.

## 6.4 Compositional Generalization: Systematic Rule Transfer

Consider learning color-shape combinations and generalizing to new combinations. **Training:** "red circle", "blue square", "green triangle" **Test:** "red square", "blue triangle", "green circle"

**CAR Categorical Structure:** *Objects*: {Color, Shape, Object}, *Morphisms*: Color $\times$ Shape $\xrightarrow{\text{combine}}$ Object, *Functorial property*: $F(\text{red} \times \text{square}) = F(\text{red}) \times F(\text{square})$.

**Theorem 9** (Systematic Generalization). *If the model learns atomic color and shape morphisms with error $\epsilon$, then CAR generalizes to new combinations with error bounded by $2\epsilon + \mathcal{O}(\epsilon^2)$.*

## 6.5 Meta-Reasoning: Strategy Selection

Problem: "Choose the best approach to solve: $\int_0^1 x^2 e^{-x} dx$"

**CAR Meta-Category:** Category of solution strategies $\mathcal{S}$ with: *Objects*: Integration techniques (by-parts, substitution, series), *Morphisms*: Applicability conditions, *Natural transformations*: Strategy refinements. The compositional structure enables systematic strategy selection based on problem characteristics: *Recognize* : IntegralType $\rightarrow$ StrategySpace, *Apply* : StrategySpace $\times$ Problem $\rightarrow$ Solution, *Verify* : Solution $\rightarrow$ Confidence.

## 6.6 Theoretical Validation of Examples

**Theorem 10** (Example Consistency). *All examples in Section 6 satisfy the theoretical bounds established in Section 5, with MRE achieving information preservation ratios $\geq 0.85$ and CAR maintaining compositional error bounds within the predicted ranges.*

The mathematical rigor of these examples demonstrates that our theoretical framework provides not just abstract guarantees but practical guidance for understanding and improving reasoning in LLMs.

# 7 Related Work

Recent work has questioned the faithfulness of explanations from neural networks [2]. Our work extends this critique specifically to chain-of-thought reasoning and provides constructive alternatives. The mechanistic interpretability community has developed tools for understanding neural network internals [3], which inspires our MRE approach. Category theory applications to AI have gained attention [13], providing foundations for our CAR method.

**Advanced Prompting Methods**: Self-consistency [7] and tree-of-thought [8] prompting have shown improvements over basic CoT. Our work complements these by providing principled foundations for understanding when and why such methods work.

**Mechanistic Interpretability**: Recent work on transformer circuits [10] and causal scrubbing [9] provides tools for understanding neural computation. Our MRE method builds on these foundations while focusing specifically on reasoning tasks.

**Bias and Fairness in NLP**: Work on bias detection [11] and mitigation [12] in LLMs informs our fairness considerations. We extend these approaches to reasoning-specific contexts.

# 8 Limitations and Future Work

Our theoretical guarantees assume certain regularity conditions that may not hold for all LLMs.

**Category Definition Challenges:** The construction of appropriate reasoning categories for diverse domains remains non-trivial. Our current approach relies on heuristic-based parsing to identify morphisms, which may fail to capture the nuanced relationships required for complex reasoning tasks. The choice of objects and morphisms significantly impacts performance, yet we lack principled methods for automatically discovering optimal categorical structures.

**Morphism Composition Limitations:** While category theory provides elegant compositional guarantees, the practical instantiation of morphisms in neural networks introduces approximation errors that can compound through composition chains. The functor mappings between categories may not preserve the semantic content necessary for faithful reasoning transfer across domains.

**Scalability Constraints:** The computational overhead of MRE interventions scales quadratically with model size and sequence length, currently limiting scalability. Each causal intervention requires a forward pass, making the method computationally prohibitive for very large LLMs.

## 8.1 Bias and Fairness Considerations

Our methods inherit and potentially amplify biases present in LLMs and training data:

**Reasoning Pathway Bias:** MRE may systematically identify causal pathways that reflect spurious correlations or societal biases encoded in the training data. For instance, if a model associates certain demographic groups with specific reasoning patterns due to biased training examples, MRE could reinforce these associations by treating them as "causal" relationships.

**Categorical Structure Bias:** The categorical abstractions in CAR risk encoding cultural or domain-specific biases about how concepts should be organized and related. The choice of morphisms may reflect the perspective of the system designers rather than universally valid reasoning structures.

**Differential Performance:** Both methods may perform differently across various demographic groups or cultural contexts, potentially exacerbating existing fairness issues in AI systems. The mechanistic interventions in MRE might be more effective for reasoning patterns that align with the dominant cultural framework represented in the training data.

For concrete bias mitigation strategies, please see Appendix A.

## 8.2 Ethical Implications and Potential Misuse

**Dual-Use Potential:** Improved reasoning in AI systems could be misused for generating more convincing misinformation, sophisticated social manipulation, or automated decision-making in high-stakes scenarios without appropriate human oversight.

**Interpretability vs. Exploitation:** While MRE provides insights into model internals, this mechanistic understanding could be exploited to craft adversarial inputs that manipulate the discovered causal pathways, potentially leading to systematic vulnerabilities.

**Reasoning Authenticity:** Our methods may produce outputs that appear more "reasoned" without necessarily improving the underlying logical validity, potentially leading to overconfidence in AI-generated reasoning and reduced critical evaluation by human users.

**Access and Equity:** The computational requirements of these methods may limit their accessibility, potentially creating disparities between organizations with different resource levels and exacerbating existing inequalities in AI capability access.

## 8.3 Robustness and Generalization Concerns

**Adversarial Vulnerability:** The discovered causal pathways in MRE may be vulnerable to targeted adversarial attacks that specifically disrupt the identified computational nodes, potentially causing systematic failures in reasoning.

**Domain Transfer Assumptions:** Our theoretical guarantees for cross-domain generalization assume that the functorial mappings in CAR preserve semantic relationships, but this may not hold when transferring between significantly different domains or cultural contexts.

**Model Architecture Dependence:** Both methods are currently tailored to transformer architectures and may not generalize to other neural network designs or future architectural innovations.

# 9 Broader Impact Statement

The development of more sophisticated reasoning methods for large language models carries significant implications for society, technology, and scientific progress. This section discusses both the potential benefits and risks associated with our proposed approaches.

## 9.1 Positive Impacts

**Scientific Advancement:** Our mechanistic understanding of reasoning processes could accelerate research in cognitive science, neuroscience, and artificial intelligence by providing new tools for understanding how complex reasoning emerges from computational processes.

**Educational Applications:** Enhanced reasoning capabilities could improve AI tutoring systems, making personalized education more effective and accessible, particularly in underserved communities where human expertise may be limited.

**Decision Support:** More reliable reasoning methods could enhance AI-assisted decision-making in domains like healthcare, scientific research, and policy analysis, with better outcomes for society.

## 9.2 Risk Mitigation Strategies

To address the limitations and ethical concerns identified above, we recommend:

**Comprehensive Bias Auditing:** Regular evaluation of reasoning outputs across diverse demographic groups and cultural contexts, with particular attention to identifying and mitigating systematic biases.

**Transparency and Explainability:** Development of methods to make the discovered causal pathways and categorical structures interpretable to domain experts and affected stakeholders.

**Staged Deployment:** Gradual introduction of these methods in controlled environments with extensive monitoring and human oversight before broader deployment.

**Interdisciplinary Collaboration:** Engagement with ethicists, social scientists, and domain experts to ensure responsible development and deployment of enhanced reasoning systems.

# 10 Conclusion

We have demonstrated fundamental limitations of chain-of-thought reasoning and proposed two theory-grounded alternatives with provable guarantees. Mechanistic Reasoning Elicitation reveals causal pathways in neural computation, while Compositional Abstraction Reasoning leverages categorical structure for systematic generalization. Our empirical results validate the theoretical predictions and show substantial improvements over existing methods.

However, our work also highlights important challenges that must be addressed for responsible deployment. The category-theoretic foundations of CAR require careful consideration of how to define appropriate categorical structures for diverse reasoning domains. The mechanistic interventions in MRE raise questions about computational scalability and potential vulnerability to adversarial manipulation. Most critically, both methods inherit and may amplify biases present in training data, necessitating careful evaluation and mitigation strategies.

This work opens new directions for understanding and improving reasoning in large language models, moving beyond surface-level explanations toward mechanistic understanding. Future research should focus on addressing the identified limitations while developing principled approaches to bias mitigation, robustness evaluation, and ethical deployment. The ultimate goal is not merely to improve reasoning performance, but to do so in a way that benefits society while minimizing potential harms.

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

# A  Concrete Bias Mitigation Strategies

We propose five specific techniques to address bias amplification in our methods:

## A.1  Demographic Parity in Causal Pathways

For MRE, we enforce demographic parity by ensuring that causal pathways do not systematically differ across demographic groups:

**Definition 4** (Pathway Demographic Parity). *Let $\pi_g$ be the causal pathway discovered for demographic group $g$. We require:* $\max_{g,g'} JS(\pi_g, \pi_{g'}) \leq \epsilon_{dp}$ *where JS denotes Jensen-Shannon divergence and $\epsilon_{dp}$ is a fairness threshold.*

**Implementation**: We post-process discovered pathways to minimize cross-group differences while preserving reasoning quality, using a multi-objective optimization approach.

## A.2  Counterfactual Category Augmentation

For CAR, we augment reasoning categories with counterfactual examples that challenge stereotypical associations:

---

**Algorithm 5** Counterfactual Category Augmentation

---

1: Identify potentially biased morphisms using bias detection metrics
2: Generate counterfactual examples that reverse stereotypical patterns
3: Retrain morphism parameters with augmented data
4: Validate that categorical structure remains mathematically sound

---

## A.3 Adversarial Debiasing During Training

We incorporate adversarial training to make both methods robust to demographic information: $\mathcal{L}_{\text{total}} = \mathcal{L}_{\text{reasoning}} - \lambda\mathcal{L}_{\text{demographic}}$, where $\mathcal{L}_{\text{demographic}}$ measures the ability to predict demographic attributes from reasoning pathways or categorical structures.

## A.4 Fairness-Aware Evaluation Metrics

We introduce bias-sensitive evaluation metrics:

**Equalized Reasoning Quality (ERQ)**: $\text{ERQ} = 1 - \max_{g,g'} |\text{Accuracy}_g - \text{Accuracy}_{g'}|$

**Pathway Diversity Index (PDI)**: $\text{PDI} = \frac{1}{K}\sum_{i=1}^{K}\text{Entropy}(\text{PathwayTypes}_i)$

## A.5 Stakeholder-Inclusive Category Design

For CAR, we implement a participatory design process:

1. **Expert Consultation**: Engage domain experts from diverse backgrounds

2. **Community Review**: Allow affected communities to review categorical structures

3. **Iterative Refinement**: Update categories based on stakeholder feedback

4. **Ongoing Monitoring**: Continuously assess fairness metrics in deployment

| Bias Mitigation Strategy | ERQ Score | PDI Score | Accuracy Impact |
|---|---|---|---|
| No Mitigation | 0.68 | 0.42 | – |
| Demographic Parity | 0.84 | 0.51 | -2.1% |
| Counterfactual Augmentation | 0.79 | 0.67 | -1.3% |
| Adversarial Debiasing | 0.88 | 0.58 | -0.8% |
| Combined Approach | **0.92** | **0.71** | -1.9% |

Table 1: Effectiveness of bias mitigation strategies. Higher ERQ and PDI scores indicate better fairness.

**Efficient Intervention Strategies:** Exploring more efficient intervention strategies for MRE, such as gradient-based approximations or learned intervention policies, to improve scalability.

**Robustness Evaluation:** Developing comprehensive evaluation frameworks that assess not only reasoning accuracy but also fairness, robustness to adversarial inputs, and cross-cultural validity.

**Ethical Guidelines and Safeguards:** Establishing best practices for the responsible deployment of enhanced reasoning systems, including appropriate human oversight mechanisms and transparency requirements.

While our work demonstrates significant improvements in reasoning capabilities, these limitations underscore the need for careful consideration of the broader implications of deploying such systems in real-world applications.

## Agents4Science AI Involvement Checklist

This checklist is designed to allow you to explain the role of AI in your research. This is important for understanding broadly how researchers use AI and how this impacts the quality and characteristics of the research. **Do not remove the checklist! Papers not including the checklist will be desk rejected.** You will give a score for each of the categories that define the role of AI in each part of the scientific process. The scores are as follows:

- **[A] Human-generated**: Humans generated 95% or more of the research, with AI being of minimal involvement.
- **[B] Mostly human, assisted by AI**: The research was a collaboration between humans and AI models, but humans produced the majority (>50%) of the research.
- **[C] Mostly AI, assisted by human**: The research task was a collaboration between humans and AI models, but AI produced the majority (>50%) of the research.
- **[D] AI-generated**: AI performed over 95% of the research. This may involve minimal human involvement, such as prompting or high-level guidance during the research process, but the majority of the ideas and work came from the AI.

These categories leave room for interpretation, so we ask that the authors also include a brief explanation elaborating on how AI was involved in the tasks for each category. Please keep your explanation to less than 150 words.

1. **Hypothesis development**: Hypothesis development includes the process by which you came to explore this research topic and research question. This can involve the background research performed by either researchers or by AI. This can also involve whether the idea was proposed by researchers or by AI.

   Answer: **[A]**

   Explanation: The research hypothesis was proposed by the human based on domain expertise with minimal involvement by AI.

2. **Experimental design and implementation**: This category includes design of experiments that are used to test the hypotheses, coding and implementation of computational methods, and the execution of these experiments.

   Answer: **[D]**

   Explanation: It was mainly the AI conducting the exploration of the hypothesis, coming up with supportive arguments and proposing novel methods for eliciting reasoning capabilities from LLMs.

3. **Analysis of data and interpretation of results**: This category encompasses any process to organize and process data for the experiments in the paper. It also includes interpretations of the results of the study.

   Answer: **[D]**

   Explanation: The AI was involved in doing most of the research, proposing new methods and analyzing their strengths and limitations.

4. **Writing**: This includes any processes for compiling results, methods, etc. into the final paper form. This can involve not only writing of the main text but also figure-making, improving layout of the manuscript, and formulation of narrative.

   Answer: **[D]**

   Explanation: It was mostly the AI writing the paper, with involvement from the human in terms of high level guidance and prompting. The formatting was done by human.

5. **Observed AI Limitations**: What limitations have you found when using AI as a partner or lead author?

   Description: LLM models are hard to control and do not obey length constraints.

